# Intestinal microbiome composition and its relation to joint pain and inflammation

Cindy G. Boer [1], Djawad Radjabzadeh[1], Carolina Medina-Gomez [1], Sanzhima Garmaeva [2], Dieuwke Schiphof [3], Pascal Arp[1], Thomas Koet[1], Alexander Kurilshikov [2], Jingyuan Fu [2,4], M. Arfan Ikram [5], Sita Bierma-Zeinstra[3,6], André G. Uitterlinden [1,5], Robert Kraaij[1], Alexandra Zhernakova[2] & Joyce B.J. van Meurs[1]*

Macrophage-mediated inflammation is thought to have a causal role in osteoarthritis-related pain and severity, and has been suggested to be triggered by endotoxins produced by the gastrointestinal microbiome. Here we investigate the relationship between joint pain and the gastrointestinal microbiome composition, and osteoarthritis-related knee pain in the Rotterdam Study; a large population based cohort study. We show that abundance of *Streptococcus* species is associated with increased knee pain, which we validate by absolute quantification of *Streptococcus* species. In addition, we replicate these results in 867 Caucasian adults of the Lifelines-DEEP study. Finally we show evidence that this association is driven by local inflammation in the knee joint. Our results indicate the microbiome is a possible therapeutic target for osteoarthritis-related knee pain.

[1] Department of Internal Medicine, Erasmus MC, University Medical Center, Rotterdam, the Netherlands. [2] Department of Genetics, University Medical Center Groningen, University of Groningen, Groningen, the Netherlands. [3] Department of General Practice, Erasmus MC, University Medical Center, Rotterdam, the Netherlands. [4] Department of pediatrics, University Medical Center Groningen, University of Groningen, Groningen, the Netherlands. [5] Department of Epidemiology, Erasmus MC, University Medical Center Rotterdam, Rotterdam, The Netherlands. [6] Department of Orthopedics, Erasmus MC, University Medical Center Rotterdam, Rotterdam, The Netherlands. *email: j.vanmeurs@erasmusmc.nl

Osteoarthritis (OA) is a degenerative joint disease and the most common form of arthritis: an estimated 22% of the adult population has at least one joint affected by osteoarthritis and this prevalence increases to 49% in individuals over 65 years of age[1]. The hallmark clinical symptom of OA is pain, which is one of the leading causes of disability in OA[2]. Pathological changes in OA affect all joint tissues: degradation of cartilage and bone, abnormal bone formation (osteophytes), and inflammation of the synovial membrane, (synovitis). Although OA is often described as predominantly caused by mechanical factors and genetic predisposition, the existence of inflammation in OA, locally or systemic, is widespread[3–5]. In addition, it has become apparent that, by promoting or exacerbating OA symptoms, predominantly OA joint pain[4–7] this local or systemic inflammation has a causal role in OA pathology[3–5].

Obesity, a well-known risk factor for OA, is thought to increase OA risk through increased mechanical loading on weight-bearing joints. However, obesity also increases the risk for OA in non-weight-bearing joints[5,8]. The increased risk of OA in non-weight-bearing joints seen in obese individuals might be directed through low-grade systemic inflammation[9,10]. The gastrointestinal microbiome has emerged as one of the factors triggering obesity associated low-grade systemic inflammation[10–13]. Obesity is associated with changes in gastrointestinal-microbiome composition, which can lead to an increased intestinal absorption of immunogenic bacterial products[12,14,15]. Gastrointestinal bacteria produce a wide range of biologically active molecules, such as metabolites, short-chain fatty acids, proteins and enzymes, of which some are secreted in, outer membrane or membrane vesicles (OMVs/MVs). All gram-negative bacteria, archaea, fungi and several gram-positive bacteria can constitutively produce these vesicles[16–18]. These vesicles are insensitive to proteases, suggesting they can transport their content over long distances from their sites of origin. The content of these vesicles can be delivered to different organs in a concentrated manner[19]. Also, some of these biologically active molecules can affect intestinal mucosal permeability (short-chain fatty acids) or activate the immune system (lipopolysaccharide)[20]. Specifically, these molecules can affect macrophage activation and Toll-like-receptor (TLR) pathways[16,21], which have recently been shown to be the predominant inflammatory responses seen in OA[4]. Indeed, elevated levels of the bacterial endotoxin LPS (lipopolysaccharide), in the blood or in the synovium of OA patients, is associated with more severe knee OA, knee pain and inflammation[22]. This and other studies link gut-microbiome composition to low-grade systemic and local inflammation seen in OA[23–25].

In the present study we examine stool microbiome as a proxy for the gastrointestinal-microbiome composition in relation to knee OA severity, OA-related knee pain, measured by the WOMAC-pain score, and obesity, in a large population-based cohort. We report a significant association between *Streptococcus* species (*spp.*) abundance in stool microbiome samples, knee WOMAC pain, knee inflammation, and replicate this finding in an independent cohort. Our results suggest a true relationship between the gastrointestinal microbiome, low-grade inflammation in the knee, and knee OA pain, independent of obesity.

## Results

**Rotterdam Study Microbiome cohort profile**. For 1427 participants from the Rotterdam Study (RSIII) we determined the gastrointestinal-microbiome composition by taking the stool microbiome as a proxy for the intestinal microbiome. In the Rotterdam Study Microbiome cohort, we sequenced two hypervariable regions of the bacterial 16S rRNA gene, hypervariable regions V3 and V4. After quality control, the 16S reads were directly mapped against the Silva 16S sequence database (v128) using RDP classifier for taxonomic classification. Classification prediction was done on multiple taxonomic levels: domain, phylum, class, order, family and genus. Gastrointestinal-microbiome composition of the 1427 participants in the Rotterdam Study Microbiome cohort is schematically presented Fig. 1. In total, there are 596 single taxonomies in our cohort, with unknown and unclassified bacteria excluded, as these could not be identified for clinical therapeutic relevance. At phylum level, the dominant phyla are *Firmucutes* (77.8%) and *Bacteriodites* (12.5%), followed by *Proteobacteria* (4.9%) and *Actinobacteria* (4.1%, Fig. 1). This is in concordance with other large-scale population-cohorts of Caucasian adults[26,27]. General characteristics of the Rotterdam Study Microbiome cohort are presented in Table 1. The study population ($n = 1427$) consisted of 57.5% women ($n = 821$) and was slightly obese, with an average body mass index (BMI) of 27.5. A total of 124 individuals had radiographic knee OA, while 285 participants reported knee OA pain (WOMAC pain score > 0). The majority of participants reporting knee pain was female ($n = 206$). The average WOMAC pain score was also significantly higher in females compared to males ($P$-value $= 1.1 \times 10^{-07}$, Student's $T$-test, Table 1).

**_Streptococcus_ abundance is associated with OA knee pain**. First, we examined whether the overall microbiome composition was different across knee WOMAC pain scores and OA severity (Kellgren-Lawrence radiographic OA severity scores). We found that knee WOMAC pain significantly contributes to the intestinal microbiome β-diversity as evaluated at genus level (Aitchison distance, $r^2 = 0.0014$, $P$-value $= 0.005$, PERMANOVA, Supplementary Fig. 1). This association was attenuated when BMI was added to the model and significance was lost ($r^2 = 0.00088$, $P$-value $= 0.143$, PERMANOVA). The intra-individual gastrointestinal-microbiome diversity (α-diversity, Shannon index and inverse Simpson) was not associated with knee WOMAC pain. For knee OA severity (KLsum, $n = 941$), we did not identify an association with gastrointestinal microbiome α- and β-diversity (Supplementary Table 1).

To gain more insight into which gut-microbiome taxonomies drive the association with knee WOMAC pain, we performed multivariate association analysis on the 256 taxonomies remaining after additional QC. After adjusting for age, sex and technical covariates, we found four microbiome abundancies significantly (FDR < 0.05) associated with knee WOMAC-pain severity (Table 2). These were all in the same clade; from class, order, family leading to the bacterial genus of *Streptococcus* (genus: coefficient $= 5.0 \times 10^{-03}$, FDR $P$-value $= 1.2 \times 10^{-05}$, MaAsLin, Table 2; See Supplementary Data 1 for the full summary statistics). After additional correction for possible confounders (smoking and alcohol consumption)[28], *Streptococcus spp.* abundance remained significantly associated with knee WOMAC pain (coefficient $= 4.8 \times 10^{-03}$, FDR $P$-value $= 3.8 \times 10^{-05}$, MaAsLin, Supplementary Table 2). This association is largely independent of BMI (coefficient $= 4.1 \times 10^{-03}$, FDR $P$-value $= 2.1 \times 10^{-03}$, MaAsLin, Supplementary Table 2). Ethnicity has recently been put forward as a possible confounder for gastrointestinal-microbiome composition[26]. However, after exclusion of individuals with non-European ancestry ($n = 163$) from our analysis, *Streptococcus spp.* remained associated with knee WOMAC pain (coefficient $= 5.0 \times 10^{-03}$, FDR $P$-value $= 5.8 \times 10^{-05}$, MaAsLin, Supplementary Table 3), also, after adjustment for confounders (coefficient $= 4.3 \times 10^{-03}$, FDR $P$-value $= 1.9 \times 10^{-03}$, MaAsLin). Altogether, we found that the gut microbiome, and in particular a greater relative abundance of *Streptococcus spp.*, is significantly associated with higher knee WOMAC pain independent of smoking, alcohol consumption and BMI.

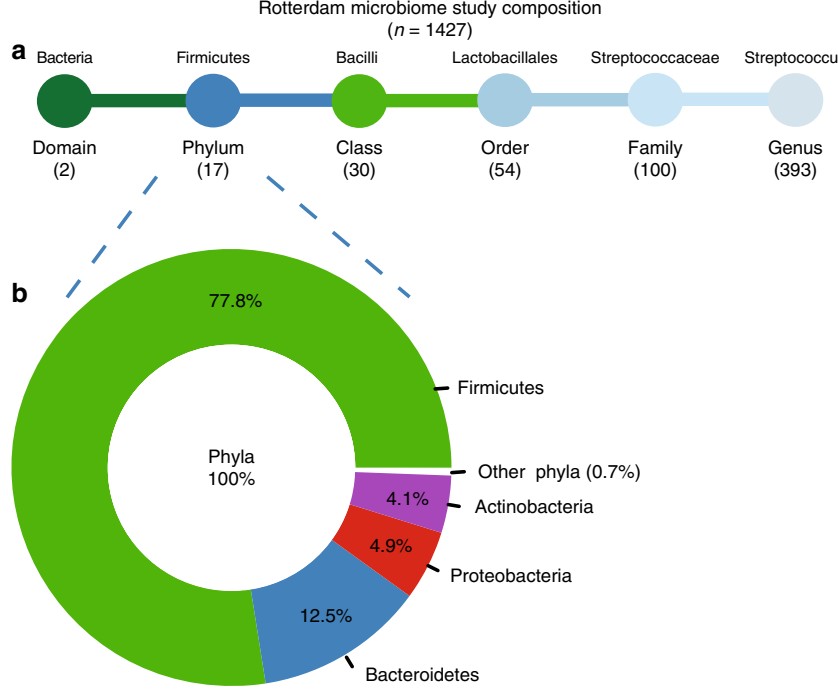

**Fig. 1** Schematic representation of the gut-microbiome taxonomic abundance in the Rotterdam Study cohort. **a** overview of the number unique taxonomies detected at each level, unknown and unclassified bacteria were excluded. **a** Above, as an example, the taxonomic classification for *Streptococcus* is shown. **b** Donut plot of the relative abundancy in percentage (%) of the different unique phyla present in the entire dataset ($n = 1427$) unknown and unclassified bacteria were excluded

| Table 1 General characteristics of the Rotterdam Study Microbiome cohort | | | |
|---|---|---|---|
| **Rotterdam Study Microbiome** | **Females** | **Males** | **Total** |
| Cohort participants | 821 | 606 | 1.427 |
| Age (years) | 56.8 (5.9) | 56.9 (5.9) | 56.9 (5.9) |
| BMI (kg/m$^2$) | 27.4 (4.9) | 27.6 (4.0) | 27.5 (4.5) |
| Alcohol (g/day) | 1.3 (2.7) | 1.3 (2.3) | 1.3 (2.6) |
| Smoking (y/n) | 98 smokers/721 non smokers | 97 smokers/507 non smokers | 195 current smokers |
| PPI (y/n) | 182 users/638 non-users | 114 users/492 non-users | 296 current PPI users |
| NSAIDs (y/n) | 127 users/693 non-users | 51 users/555 non-users | 178 current NSAID users |
| Knee phenotypes | | | |
| Knee OA (y/n) | 84 cases/456 controls | 40 cases/361 controls | 124 cases/817 controls |
| KLSum score | 1.0 (1.4) | 0.7 (1.2) | 0.8 (1.3) |
| WOMAC-Pain score | 1.2 (2.6) | 0.6 (1.9) | 0.9 (2.3) |
| WOMAC-Pain score > 0 | 206 | 79 | 285 |
| α-diversity metrics | | | |
| Shannon Index | 4.0 (4.1) | 4.0 (4.0) | 4.0 (0.5) |
| Inverse Simpson Index | 26.0 (12.1) | 25.5 (12.2) | 25.8 (12.2) |

Depicted are the mean and the SD (standard deviation) in parenthesis
*PPI* oral use of proton pump inhibitors, *NSAIDs* oral use of non-steroidal anti-inflammatory drugs, *OA* osteoarthritis, *WOMAC* Western Ontario and McMaster Osteoarthritis Index

***Streptococcus* association not driven by oral medication use**. Proton Pump Inhibitors (PPI) are among the most widely used over-the-counter drugs in the world. They are used to treat gastro-esophageal reflux and prevent gastric-ulcers. Recent research has shown that the gastrointestinal-microbiome composition of PPI users is profoundly different from non-PPI users, mainly due to a strong increase in *Lactobacilli* abundance, driven by *Streptococcus spp.*[29]. Another potential over-the-counter medication frequently taken by patients with joint complaints are non-steroidal anti-inflammatory drugs (NSAIDs), which also affect the gastrointestinal microbiome[29]. To investigate, whether the association between *Streptococcus spp.* abundance and knee

WOMAC pain is mediated by use of these drugs, we included PPI and NSAID use as covariates in our model (in addition to age, sex, technical covariates, smoking, alcohol consumption, and BMI). After adjustment for these covariates, the coefficient was attenuated, but the association remains significant (coefficient = $3.4 \times 10^{-03}$, P-value = $2.4 \times 10^{-04}$, MaAsLin, Supplementary Table 4). In line with these results, excluding all current PPI users ($n = 265$) from the analysis, greater *Streptococcus spp.* abundance was significantly associated with higher knee WOMAC pain ($n = 1104$, coefficient = $2.4 \times 10^{-03}$, P-value = $4.0 \times 10^{-03}$, MaAsLin). Thus, the association between, *Streptococcus spp.* and knee WOMAC pain was not due to the confounding effect of PPI use.

**Table 2 Results of multivariate linear regression analysis of gut microbiome relative abundancies and knee WOMAC-pain scores, in the Rotterdam Study, LifeLines-DEEP**

| Taxonomy | % | RS N | RS CoE | RS SE | RS P-value | RS FDR | LLD N | LLD CoE | LLD SE | LLD P-value | Meta N | Meta P-value |
|---|---|---|---|---|---|---|---|---|---|---|---|---|
| Class *Bacilli* | 27.3% | 1419 | $6.1 \times 10^{-03}$ | $1.0 \times 10^{-03}$ | $9.1 \times 10^{-09}$ | $1.0 \times 10^{-06}$ | 867 | $5.4 \times 10^{-03}$ | $1.3 \times 10^{-03}$ | $3.6 \times 10^{-05}$ | 2286 | $1.1 \times 10^{-12}$ |
| Order *Lactobacillales* | 100% | 1417 | $6.1 \times 10^{-03}$ | $1.1 \times 10^{-03}$ | $7.6 \times 10^{-09}$ | $1.0 \times 10^{-06}$ | 864 | $4.9 \times 10^{-03}$ | $1.3 \times 10^{-03}$ | $2.4 \times 10^{-04}$ | 2281 | $8.3 \times 10^{-12}$ |
| Family *Streptococcaceae* | 79.6% | 1402 | $4.9 \times 10^{-03}$ | $9.3 \times 10^{-04}$ | $1.5 \times 10^{-07}$ | $8.7 \times 10^{-06}$ | 863 | $2.9 \times 10^{-03}$ | $1.3 \times 10^{-03}$ | $2.3 \times 10^{-02}$ | 2265 | $2.1 \times 10^{-08}$ |
| Genus *Streptococcus* | 98.7% | 1396 | $5.0 \times 10^{-03}$ | $9.3 \times 10^{-04}$ | $7.3 \times 10^{-08}$ | $5.6 \times 10^{-06}$ | 860 | $3.3 \times 10^{-03}$ | $1.6 \times 10^{-03}$ | $3.7 \times 10^{-02}$ | 2256 | $1.3 \times 10^{-08}$ |

Adjusted for age, sex and, technical covariates: DNA isolation batch and TimeInMail. RS: Rotterdam Study (n = 1427) P-value, CoE and SE from MaAsLin analysis, LLD: LifeLines-DEEP (n = 867) P-value, CoE and SE from MaAsLin, Meta: Rotterdam Study and LifeLines Deep meta-analyzed together, sample size weighted inverse-variance meta-analysis in METAL. Taxonomy% = percentage of taxonomy is from one taxonomy level higher, ex. 23.7% of all Firmicutes are Bacilli. N = number of individuals in cohort where microbial abundance is not zero for that taxonomy. FDR: P-value adjusted for multiple testing, Benjamin-Hochberg false discovery rate. CoE coefficient, SE standard error

Since knee WOMAC-pain scores were not normally distributed (left-skewed with an overabundance of zeros), we also assessed the association using a Poisson-regression model, which can account for the non-normal distribution. Here as well, a significant association with *Streptococcus spp.* abundance was observed (beta = 1.85, P-value = $1.4 \times 10^{-04}$, Poisson-regression). Further sensitivity analysis excluding individuals who did not report pain (n = 1142) or whose knee WOMAC-pain scores were deemed as outliers (WOMAC-pain score >10, > 4 SD, n = 56) resulted in slight changes in the association coefficients. The association, however, remained significant (Supplementary Table 5). Altogether, these analyses show that the association between relative abundance of *streptococcus spp.* and knee WOMAC pain is robust.

**Quantitative determination of *Streptococcus spp*.** As 16S rRNA-sequencing derived relative microbiome data cannot provide information about the extent or directionality of changes in microbiome taxa abundance[30], we determined the absolute amount of *Streptococcus spp.* in the individuals of our study population. For each sample in our cohort (n = 1427), we quantified the number of *Streptococcus spp.* using genus specific qPCR and the total microbial load using 16S rRNA qPCR. We calculated the absolute quantity of *Streptococcus spp.* and normalized for the total bacterial load in each samples as measured by 16S rRNA qPCR. The 16S rRNA-sequencing results and qPCR *Streptococcus spp.* quantity yielded similar results (Spearman correlation, $r = 0.80$, P-value = $2.2 \times 10^{-16}$, Supplementary Fig. 2). Using the absolute abundance of *Streptococcus* measured by qPCR instead of the relative abundance derived from the 16S rRNA-sequencing profiles, we again found a significant association between higher knee WOMAC pain and greater absolute *Streptococcus spp.* abundance (beta = 0.10, P-value = $7.4 \times 10^{-03}$, Poisson regression), also after adjustment for smoking, alcohol consumption, and BMI (beta = 0.074, P-value = $4.5 \times 10^{-02}$, Poisson regression).

To adjust for possible spurious collinearity between microbe abundances[31], we have used the isometric log-ratio transformation (ILR). Using ILR, we have compared the relative *Streptococcus spp.* abundancy against the geometric mean of the abundancy of all other genera. Results show that the ILR transformed *Streptococcus spp.* abundancy is associated with knee WOMAC pain (P-value = $9.9 \times 10^{-06}$, MaAsLin, Supplementary Table 6). The association remained significant after adjusting for smoking, alcohol consumption and BMI (P-value = $8.4 \times 10^{-04}$, MaAsLin, Supplementary Table 6), and NSAID and PPI use (P-value = $8.5 \times 10^{-03}$, MaAslin, Supplementary Table 6). The qPCR results and ILR transformation demonstrate that the association between *Streptococcus spp.* abundance and knee WOMAC pain is not an artefact of the 16S rRNA sequencing.

**Independent replication of *Streptococcus* association.** We sought replication for all four associations with WOMAC pain, i.e., class, order, family and, the bacterial genus of *Streptococcus*, in an independent Dutch cohort, LifeLines-DEEP (LLD) (Supplementary Table 7). LLD has a lower sample size (n = 867), younger age (mean age = 45.6) and fewer individuals with OA-related knee pain (WOMAC pain > 0, n = 197), however, the average knee WOMAC pain was very similar compared with the Rotterdam Study cohort (RS = 0.9 and LLD = 0.9, Table 1 and Supplementary Table 7). Despite lower power in the replication cohort we observed a significant association (P-value < 0.05) between knee WOMAC-pain scores and all four taxonomies to the genus of *Streptococcus* (coefficient$_{replication}$ = $3.3 \times 10^{-03}$, P-value$_{replication}$ = $3.7 \times 10^{-02}$, MaAsLin, Table 2). Also, in the

**Table 3 Results of the association analysis of *Streptococcus* and knee joint effusion**

| Taxonomy | N | Model 1 CoE | Model 1 P-value | Model 2 CoE | Model 2 P-value |
|---|---|---|---|---|---|
| Class *Bacilli* | 314 | $9.4 \times 10^{-03}$ | $3.4 \times 10^{-02}$ | $2.7 \times 10^{-03}$ | $3.5 \times 10^{-01}$ |
| Order *Lactobacillales* | 314 | $9.8 \times 10^{-03}$ | $2.7 \times 10^{-02}$ | $2.7 \times 10^{-03}$ | $3.6 \times 10^{-01}$ |
| Family *Streptococcaceae* | 310 | $9.6 \times 10^{-03}$ | $1.7 \times 10^{-02}$ | $3.0 \times 10^{-03}$ | $2.6 \times 10^{-01}$ |
| Genus *Streptococcus* | 308 | $1.0 \times 10^{-02}$ | $1.3 \times 10^{-02}$ | $3.3 \times 10^{-03}$ | $2.1 \times 10^{-01}$ |

Knee joint inflammation was measured as severity of effusion as measured on knee MRI. Knee MRI's were only available for an all-female obese subgroup of the Rotterdam Study Microbiome dataset ($n = 373$). First model assessed the association of Knee effusion with the microbiome, adjusted for age, sex, DNA isolation batch and TimeInMail (technical covariates). Second model was WOMAC-pain score adjusted for age, sex, technical covariates and, effusion severity. *P*-values were determined by MaAsLin analysis. N = number of individuals in cohort where microbial abundancy is not zero for that taxonomy
*CoE* coefficient

meta-analysis of RS and LLD, greater *Streptococcus spp.* abundance was significantly associated with higher knee WOMAC pain (*P*-value$_{meta}$ = $1.3 \times 10^{-08}$, MaAsLin, $n = 2256$, Table 2). After adjusting for BMI, we found significant replication on class, order and the family level of *Streptococcacea* (coefficient$_{replication}$ = $2.7 \times 10^{-3}$, *P*-value$_{replication}$ = $3.5 \times 10^{-02}$, MaAsLin, Supplementary Table 8). In the replication study, additional adjustment for BMI slightly attenuated the association, a 9.1% decrease in coefficient, for the genus of *Streptococcus* (Supplementary Table 8). This was in concordance with the 14.5% decrease in coefficient for *streptococcus*, seen in the discovery cohort after BMI adjustment (Supplementary Table 2). In the BMI adjusted meta-analysis for RS and LLD, the association between knee WOMAC pain and *Streptococcus spp.* was highly significant (*P*-value$_{meta}$ = $1.1 \times 10^{-06}$, METAL, $n = 2256$, Supplementary Table 8).

***Streptococcus* association with knee joint inflammation**. If *Streptococcus spp.* abundance is causally related to higher knee WOMAC pain, a possible mechanism might be through local priming of macrophages in the synovial lining resulting in inflammation[22]. For a random subset of the Rotterdam Study Microbiome cohort, knee magnetic resonance images (MRI) were available ($n = 373$, all females)[32] at the time of microbiome measurements. Knee joint inflammation was assessed by scoring the amount of effusion in the tibiofemoral joint and in the patellofemoral joint for both knees. We found that higher WOMAC-pain scores were significantly correlated with more knee effusion (Spearman correlation $r = 0.14$, *P*-value = $9.0 \times 10^{-03}$). In addition, we observed that greater *Streptococcus spp.* abundance was significantly associated with more effusion in the knee joints (coefficient = $1.0 \times 10^{-02}$, *P*-value = $1.3 \times 10^{-02}$, MaAsLin, Table 3). When WOMAC pain was added to the model, the association of knee effusion with *Streptococcus spp.* disappears (coefficient = $3.3 \times 10^{-03}$, *P*-value = 0.21, MaAsLin), indicating that the association of *Streptococcus spp.* with knee pain severity is driven by knee inflammation severity.

## Discussion

Using a large, deeply phenotyped population-based cohort, we identified a significant association between greater relative and absolute *Streptococcus spp.* abundance and higher OA-related knee pain. These results were validated by replication in an independent cohort and by meta-analysis. Finally, we presented evidence that this association was driven by local inflammation in the joint.

We observed that the intestinal microbiome β-diversity was significantly associated with knee WOMAC scores. After testing 256 taxonomies individually, we found a microbiome-wide association with knee WOMAC pain and Streptococcus *spp.*, where greater *Streptococcus spp.* relative abundance is associated with higher knee WOMAC pain. We found this association to be

robust, not be caused by outlier observations, or due to the confounding effects of smoking, alcohol intake, oral medication usage or BMI[28,29]. Neither was the association an artefact of the microbiome profiles as relative fractions of the 16S rRNA sequencing, or due to possible co-linearity in the data[30,33].

Although, the results of the sensitivity analyses were in line with a true association between *Streptococcus spp.* and knee WOMAC pain in our cohort, validation in an independent cohort is essential. Replication of microbiome abundance, however, is difficult, because data might not be similar between studies if sample preparation and data analysis are not done in the exact same way[34]. We sought replication in the LLD cohort, which has a different study population and data preparation method than our cohort does[27]. Despite these differences, we could replicate our association in LLD for all taxonomic levels, class, order, family and for the genus of *Streptococcus*. Also, in the meta-analysis of RS and LLD, *Streptococcus spp.* was significantly associated with knee WOMAC pain.

Obesity-mediated gastrointestinal-microbiome changes are postulated to affect low-grade systemic and local inflammation in OA[23–25]. Nevertheless, in our study the effect of *Streptococcus spp.* on knee WOMAC pain is not fully driven by BMI. This suggests a direct role for the gastrointestinal microbiome in OA-related knee pain and inflammation. We postulate that greater *Streptococcus spp.* abundance leads to higher knee WOMAC pain through local joint inflammation. This is in line with our observation that *Streptococcus spp.* abundance was significantly associated with effusion severity in the knee joints. This leads to believe that *Streptococcus spp.* might also be involved in other inflammatory joint pain disorders. This is not unlikely since several *Streptococcus spp.* have been linked to osteomyelitis[35,36], rheumatic fever[37,38] and, post-streptococcal reactive arthritis[39,40]. The last two are disorders in which due to molecular mimicry with group A *Streptococcus*, cross-reactive antibodies are produced against joint tissues, leading to rheumatic joint inflammation and damage[38]. However, these disorders involve mainly pathogenic species, such as *S. pyogenes*. Yet, most *Streptococcus spp.* are commensal species and have been found throughout the human oral-gastro-intestinal microbiome[41–43], still, these can produce immunogenic bacterial products. Several *Streptococcus spp.* have been shown to constitutively produce MVs[18]. These MVs may present *Streptococcus spp.* epitopes and/or may contain immunogenic products[17,24]. Such bacterial products can trigger macrophage activation through TLR pathways. This type of macrophage activation is predominantly seen in OA-related joint inflammation[16,21] and is thought to be related to pain[3–7].

We therefore propose that greater *Streptococcus spp.* abundance may lead to an increase of bacterial products in the circulation through increased production of metabolites that pass the gut-blood barrier or through immunogenic products that prime local or systemic macrophages. We have summarized this hypothetical model in Fig. 2.

**Fig. 2** Proposed hypothetical me pathophysiological mechanism explaining the association between the gut microbiome (*Streptococcus spp.*), knee WOMAC pain and knee effusion. No causality has been established between *Streptococcus spp.* abundance and OA-related knee pain, however, if such causality exists, we propose the following model: Members of the *Streptococcus spp.* are known to produce metabolites and membrane vesicles, which both may interact with host cells. These bacterial products can pass the gut-blood barrier, and possibly either **a** target the knee joint through activation of macrophages in the synovial lining, leading to joint inflammation and damage, or **b** enter the circulation, activate macrophages to pro-inflammatory macrophages, which may trigger a low-grade systemic inflammatory state, invoking or exacerbate joint inflammation and damage, leading to increased knee pain

Our results show a difference in microbiome β-diversity in individuals with higher knee WOMAC-pain scores, which is driven by greater abundancy of *Streptococcus spp.* This association is highly robust. Moreover, we replicated our findings in an independent cohort. However, our study has some limitations. First, a cross-sectional study design was used and cannot unequivocally establish causality. For this, longitudinal studies are required. Second, other follow-up studies are needed to better elucidate the molecular pathway connecting *Streptococcus spp.*, knee inflammation and WOMAC pain, to validate or reject our postulated hypothesis (Fig. 2). Blood and joint tissue could be examined for the presence of *Streptococcus* MVs, metabolites, and their possible association to knee inflammation and WOMAC pain severity. Third, to examine whether the association found in this study is unique for knee OA pain, other quantitative pain measurements should be examined, as well as measurements of pain at other joints. In, addition, other inflammatory joint disorders could also be examined. Forth, due to the limited resolution of 16S rRNA-sequencing methodology, we were unable to identify whether a specific *Streptococcus* species or strain was driving the association. Last, we find no association between knee OA severity (KLsum) and gastrointestinal-microbiome composition. Knee OA severity, however, was measured by radiographic OA severity, and this does not include the clinical symptom of joint pain. Although considering our proposed mechanism, an effect on knee OA severity could be expected, as inflammation can lead to joint damage. It is possible that our study currently lacks the power to detect such an association, or requires a longitudinal design to detect such effects on OA severity or disease progression.

In sum, we demonstrate an association between greater abundance of *Streptococcus spp.* and higher osteoarthritis-related knee pain, but the causality of this association needs to be established. A possible explanation for the found association is the induction or exacerbating of local joint inflammation by *Streptococcus spp.* The precise mechanisms by which

*Streptococcus spp.* may trigger joint inflammation, are not known. We hypothesized that it may involve metabolites or MVs produced by *Streptococcus spp.* in the gastrointestinal tract. As joint pain is one of the most impacting clinical OA symptoms and pain in OA has a high socioeconomic burden, it is of crucial importance to identify novel treatments that reduce clinical symptoms, in particular pain, and decrease the socioeconomic burden. Our results point to the microbiome as a therapeutic target for OA-related joint pain and possibly for other inflammatory joint pain disorders. The gastrointestinal microbiome is a promising therapeutic target, because it is sensitive to change through (diet) interventions. This could offer an easily accessible and safe treatment options for OA associated joint pain. Therefore, it is pivotal to explore the causal mechanism and possible translation of the present findings into clinical practice.

## Methods

**Rotterdam study cohort.** The Rotterdam study (RS) is a large population-based prospective study population (14,926 participants) ongoing since 1990 to study determinants of chronic disabling diseases. The first cohort RS-I, started in 1990 and includes individuals of 55 years and older living in the Ommoord district of Rotterdam in the Netherlands. In 2000, a second cohort was started RS-II with individuals who had become 55 years of age or moved into the study district since the start of the study. In 2006, the third cohort was initiated, RSIII, of individuals aged > 45, living in the Ommoord district. The cohorts are predominantly Caucasian and a further detailed description of the design and rationale of the Rotterdam Study has been published elsewhere[44]. The Rotterdam Study has been approved by the Medical Ethical Committee of the Erasmus MC, University Medical Center Rotterdam, the Netherlands (MEC 02.1015). All subjects provided written consent prior to participation in the Rotterdam Study.

Stool sample collection started in 2012 during the second visit of the Rotterdam Study III population (RSIII-2). A random group of 2440 participants was invited to provide a stool sample. In total 1691 (response rate = 69%) stool samples were received via mail at the Erasmus MC for analysis of stool microbiome composition as a marker for gastrointestinal-microbiome composition. After quality control 1427 samples remained for further analysis.

**Taxonomic profiling of gastrointestinal microbiota in RS.** Sample collection: Stool samples were collected at home by the participant using a Commode Specimen Collection System (Covidien, Mansfield, MA) and ~1 g aliquot was transferred to a 25 × 76 mm feces collection tube, which was sent via the regular mail to the Erasmus MC. Participants also filled out a short questionnaire addressing date and time of defecation, current or recent antibiotics use, current probiotics use, and recent travel activities. Upon arrival samples were stored (−20 °C), samples taking longer than 3 days to arrive at the Erasmus MC were excluded from further analysis, for the remaining samples TimeInMail (in days) was registered as a technical covariate. DNA isolation: For each participant, frozen stool samples were allowed to thaw for 10 min at room temperature prior to DNA isolation. An Aliquot of ~300 mg of stool was homogenized with 0.1 mm silica beads (MP Biomedicals, LLC, Bio Connect Life Sciences BV, Huissen, The Netherlands) and DNA was isolated from the samples using the Arrow stool DNA kit according to the manufacturers' protocol (Arrow Stool DNA; Isogen Life Science, de Meern, The Netherlands). 16S rRNA gene sequencing: The V3 and V4 hypervariable regions of the 16S rRNA gene were amplified using the 319F (ACTCCTACGGGAGGCA GCAG) −806R (GGACTACHVGGGTWTCTAAT) primer pair and dual indexing[45], a full list of all primers used can be found in Supplementary Table 9. Amplicons were normalized using the SequalPrep Normalization Plate kit (Thermo Fischer Scientific) and pooled. Amplicon pools were purified prior to sequencing (Agencourt AMPure XP, Beckman Coulter Life Science, Indianapolis, IN) and size and quantity was assessed (LabChip GX, PerkinElmer Inc., Groningen, The Netherlands). A Control library was added to ~10% of each amplicon pool as a positive control (PhiX Control v3 library, Illumina Inc., San Diego, CA). The hypervariable V3 and V4 regions of the 16S rRNA gene were sequenced in paired-end mode (2 × 300 bp) using the MiSeq platform (Illumina Inc., San Diego, CA) with an average depth of 50,000 paired reads per sample. Data processing and quality control: Sequence read quality control and taxonomic classification, was done using an in-house pipeline (μRAPtor) based on QIIME version 1.9.0[46] and UPARSE version 8.1[47]. Low-quality, merged, and chimeric reads were excluded. Duplicate samples, samples with <10,000 reads, and samples from participants that have used antibiotics (self-reported) in the 6 months prior to sample production were excluded. The remainder of the reads (~93%) were normalized using random 10,000 read subsampling (rarefication). To reconstruct taxonomic composition a direct classification of 16S sequencing reads using RDP classifier (2.12) and the SILVA 16S rRNA database (relase.128)[48]. This was done for each taxonomic level available: domain, phylum, class, order, family, and genus, with binning posterior probability cutoff of 0.8. The microbial Shannon diversity index was calculated on

the taxonomic level of genera, using vegan package in R. Relative abundances were calculated for each taxonomic level prior to any additional QC (domain, phylum, class, order, family and genus). Unknown or unassigned classifications were excluded from the final dataset (n = 69), as these currently cannot be used for clinical or therapeutic applications. In total, we were left with gut-microbiome taxonomies for 1427 individuals and 596 taxonomic classifications for analysis (Fig. 1).

**Phenotype descriptions.** Osteoarthritic knee joint pain was determined by the WOMAC questionnaire, which is a disease specific questionnaire to assess the severity of hip and knee OA, consisting of 24 items covering three domains: pain, stiffness, and function[49]. The WOMAC includes five items that measure OA joint specific pain, scored on a 5-point Likert scale, 0–4. The five knee specific WOMAC-pain scores were summed to create the knee WOMAC pain score, ranging from 0 to 20, where higher scores represent worse OA-related knee joint pain. Knee OA severity was determined by the radiographic Kellgren and Lawrence score (KL-score)[50]. Using radiographs of both knee joints, left and right, the KL-score was determined for each joint. Scores were subsequently summed for the left and right knee to form the Knee KLsum score. Knee joint effusion could be determined, for an all-female random subset of the Rotterdam Study III cohort (RSIII), by knee MRIs using a 1.5T MRI scanner (General Electric Healthcare, Milwaukee, Wisconsin, USA). For further detail of our used MRI protocol see ref. [51]. All MRI images were scored by a trained reader (blinded for clinical, radiographic and genetic data). Joint effusion was determined in the tibiofemoral joint (TFJ) and in the patellofemoral joint (PFJ) together (grade 0–3: 0 = no joint effusion, 1 = small joint effusion, 2 = moderate joint effusion, and 3 = severe joint effusion). The scores of the left and the right knee were summed, resulting in a score from 0 to 6, where higher scores represent more severe knee joint effusion. For all phenotypes, if data on either the left or right knee joint was missing, individuals were excluded. Oral medication usage such as, proton pump inhibitors (PPIs) and non-steroidal anti-inflammatory drugs (NSAIDs), were determined by questionnaire at the same time point as WOMAC scores, knee X-rays, knee MRIs, and stool sample collection.

**Lifelines-DEEP replication cohort.** The LifeLines-DEEP (LLD) cohort is a sub-cohort of the LifeLines cohort (167,729 participants) that employs a broad range of investigative procedures to assess the biomedical, socio-demographic, behavioral, physical and psychological factors that contribute to health and disease in the general Dutch population[52]. A subset of approximately 1500 participants was included in LLD subcohort. For these participants, additional biological materials were collected, including analysis of the gut-microbiome composition. The collection, phenotyping, and processing of LLD have been described in detail[27,53]. Briefly, microbiome data was generated for 1179 LLD samples. Fecal samples were collected at home within two weeks of blood sample collection and stored immediately at 20 °C. After transport on dry ice, fecal samples were stored at −80 °C. Aliquots were made and DNA was isolated with the AllPrep DNA/RNA Mini Kit (Qiagen; cat. #80204). The 16S rRNA gene of the isolated DNA was sequenced at the Broad Institute, Boston, using Illumina MiSeq pair-ends. Hypervariable region V4 was selected using forward primer 515F (GTGCCAGCM GCCGCGGTAA) and reverse primer 806 R (GGACTACHVGGGTWTCTAAT) (Supplementary Table 9)[54]. Closed-reference OUT picking has been done with 97% similarity cutoff using UCLUST[55] program and GreenGenes 13.5 reference database[56] from QIIME1[46] software. Overall, for 878 samples, both WOMAC scores and microbiome information was available. The library size of microbial sequencing was rarefied to 10,000 read-depth using the rarefy function in R package vegan (version 2.5–2). At this depth, 11 subjects were excluded. After the exclusion step, we had 867 samples (362 men and 505 women) remained for the final analysis. Their characteristics are summarized in Supplementary Table 7. The LifeLines-DEEP study has been approved by the medical ethical committee of the University Medical Center Groningen, The Netherlands.

**qPCR replication of 16S rRNA-sequencing results.** To validate the 16S rRNA-sequencing results we determined the absolute quantitative amount of *Streptococcus spp.* using qPCR. We determined the amount of *Streptococcus spp.* for each fecal DNA sample using the BactoReal qPCR assay (Ingenetix GmbH, Vienna, Austria) based on the *Streptococcus* 23S rRNA gene. A standard curve of a Plasmid standard containing the 23S rRNA gene (Ingenetix GmbH, Vienna, Austria) was included in each plate to calculate the amount of *Streptococcus spp.* present in each sample. Each sample was run in duplo in a 384-wells PCR plate containing 40 ng fecal DNA, 1x primers and probes (Ingenetix GmbH, Vienna, Austria), 1x TaqMan Gene Expression Master Mix (Life technologies) in a total volume of 5 μl. The qPCR was performed in a QuantStudio 7 Flex (ThermoFisher) with an initial denaturation at 95 °C for 10 min, followed by 40 cycles of 95 °C for 15 s and 60 °C for 60 s. To normalize the amount of *Streptococcus spp.* for the total amount of bacteria in each sample, a 16 S qPCR was performed (U16SRT-F: ACTCCT ACGGGAGGCAGCAGT and U16SRT-R: TATTACCGCGGCTGCTGGC, Supplementary Table 9)[57]. A standard curve of a Bacterial DNA Standard (Zymo Research) sample was included in each PCR plate to calculate the total amount of bacteria. Each sample was run in duplo in a 384-well plate containing 200 pg fecal

DNA, 200 pmol forward and reverse primers, 1x SYBR Fast ABI PrismTM Mastermix (KapaBiosystems) in a total volume of 5 µl. The qPCR was performed in a QuantStudio 7 Flex (ThermoFisher) with an initial denaturation at 95 °C for 3 min, followed by 40 cycles of 95 °C for 5 s and 60 °C for 20 s. Absolute abundancies were calculated from the standard curves. We adjusted the total absolute abundancy of bacteria for the average number of 16S copies (4.2 copies per bacteria[58]). We adjusted the absolute abundance of *Streptococcus spp.* for the average number of 23S rRNA gene copies in *Streptococcus spp.* (3.36 copies per *Streptococcus spp.*, based on *S. pneumoniae, S. thermophiles, S. pyogens, S. parasanguinis, S. dysgalactiae, S. salivarius, S. suis, S. mutans,* and *S. agalactiae*). We normalized the absolute abundance of *Streptococcus spp.* by calculating the log transformed value of the number of *Streptococcus spp.* per 1000 bacteria in each sample:

$$\log\left(\frac{Streptococcus\ absolute\ abundance}{total\ abundance\ of\ bacteria/1000}\right)$$

**Statistical analysis**. Statistical analyses were performed in *R: A Language and Environment for Statistical Computing*[59]. Inter-individual microbial composition (β-diversity) was calculated using the Aitchison distance calculated from the CLR (centered log-ratio) normalized data. CLR normalization was calculated on the counts of the directly taxonomic classified reads. The full dataset, including unknown and unclassified taxonomies was used for the CLR. To all read counts 1 was added to cope with the overabundance of zero's in the data, for CLR[33]. Subsequent statistical analysis was done through permutation analysis of variance (PERMANOVA) to inspect the global effect of WOMAC-pain score and KLsum score on the overall microbiome profiles, using the *adonis* PERMANOVA function in VEGAN. PERMANOVA models included age, sex, TimeInMail, DNA isolation batch, BMI and WOMAC-pain score or KLsum, in that order. Results were visualized using a PCA plot. CLR method and PCA plot were based on Gloor et al.[33]. Intra-individual microbial composition metrics (α-diversity) used are the Shannon Index and Inverse Simpson Index. Association of α-diversity with WOMAC pain score was examined by Poisson-regression model adjusted for age, sex, batch and TimeInMail. To identify associated microbiological taxa with the investigated phenotypes, we have used the multivariate statistical linear regression analysis, R package: MaAsLin[60]. MaAsLin is a specialized statistical R package for the analysis of microbial community abundance data and clinical/phenotypic metadata. All unknown and unclassified taxonomies were excluded from this analysis (*n* = 63). In MaAsLin, we have used adjusted default settings, i.e., Linear model based, quality control (QC) and exclusion of outliers based on the Grubbs test on the microbiome data only, and arcsine-square-root transformation of the single taxonomies relative abundance table to normalize the microbiome profiles. After MaAsLin QC we were left with 256 taxonomies for analysis (2 domains, 12 phyla, 18 classes, 25 orders, 41 families, and 158 genera). Missing values were not imputed, nor was automatic QC of the metadata, WOMAC pain score and covariates, or boosting by excluding metadata from the association analyses performed. For each analysis we forced the following cofactors: age (years), sex (0/1), technical covariates: DNA isolation batch (0/1) and TimeInMail (days). Depending on the model we additionally forced the following cofactors: BMI (body mass Index), smoking (current smoker, y/n), daily alcohol consumption (glass/day), PPI use (y/n), and NSAID use (y/n). Statistical significance was determined by multiple testing correction, Benjamini-Hochberg False discovery rate (FDR) < 0.05. Removal of possible collinearity in the microbiome data was done by ILR (isometric log-ratio transformation) on the counts of the directly taxonomic classified reads. The full dataset including unknown and unclassified taxonomies was used. Multivariate linear regression model adjusted for age, sex, and cohort-specific technical covariates was performed on the ILR transformed data. Replication analysis in LifeLines-DEEP (LLD), used MaAsLin using similar settings, with the exception of the automatic QC in MaAsLin. In LLD the QC of the metadata was done manually. All analysis were also adjusted for age, sex, and cohort-specific technical covariates. Meta-analysis of RS and Lifelines was performed using inverse-variance weighting by METAL[61]. Correlation between 16S sequencing *Streptococcus spp.* abundancy and qPCR *Streptococcus spp.* abundance was done by Spearman correlation in R. Association of knee WOMAC pain and qPCR data were done by Poisson-regression models adjusted for age, sex, and qPCR technical covariates (plate number). All figures and graphs were made in R and adapted in Adobe Illustrator.

**Reporting summary**. Further information on research design is available in the Nature Research Reporting Summary linked to this article.

## Data availability

All relevant data supporting the key findings of this study are available within the article and its supplementary information files. Data underlying Fig. 1 and Supplementary figures 1 and 2 are provided as Source Data file. Other data are available from the corresponding author upon reasonable requests. Due to ethical and legal restrictions, individual-level data of the Rotterdam Study cannot be made publicly available. Data are available upon request to the data manager of the Rotterdam Study Frank van Rooij (f.vanrooij@erasmusmc.nl) and subject to local rules and regulations. This includes submitting a proposal to the management team of RS, where upon approval, analysis needs to be done on a local server with protected access, complying with GDPR regulations.

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

## Acknowledgements

The Rotterdam Study is funded by Erasmus Medical Center and Erasmus University, Rotterdam, Netherlands Organization for the Health Research and Development (ZonMw), the Research Institute for Diseases in the Elderly (RIDE), the Ministry of Education, Culture and Science, the Ministry for Health, Welfare and Sports, the European Commission (DG XII), and the Municipality of Rotterdam. We are grateful to the study participants, the staff from the Rotterdam Study and the participating general practitioners and pharmacists. We wish to thank John P. Hays and Stefan Broers, for their input and discussion on the qPCR experiments and for kindly providing *Streptococcus* DNA material for testing. The generation and management of gut-microbiome data for the Rotterdam Study (RSIII) was executed by the Human Genomics Facility (HuGe-F) of the Genetic Laboratory of the Department of Internal Medicine, Erasmus MC, Rotterdam, The Netherlands. We want to thank Nahid El Faquir, Jolande Verkroost, Pelle van der Wal, Hafsa Amanat, Kamal Arabe, Hedayat Razawy, and Karan Sing Asra, for their help in sample collection, registration, and DNA isolation and sequencing. Last, we thank the Erasmus Postgraduate School Molecular Medicine (MolMed or MM) of the Faculty of Medicine and Health of the Erasmus University of Rotterdam. We would like to thank Gaby M. van Dijk for proof reading of the paper. This study was funded by The Netherlands Society for Scientific Research (NWO) VIDI Grant 917103521. D.R. was funded by an Erasmus MC mRACE grant "Profiling of the human gut microbiome". C.M.G. is partially funded by the NWO-VIDI 016.136.367. A.Z. holds an ERC starting grant (715772), and NWO-VIDI grant 016.178.056. A.Z. and J.F. are funded by Cardi-oVasculair Onderzoek Nederland (CVON 2012-03). J.F. is funded by an NWO-VIDI grant 864.13.013. S.G. holds scholarship from the Graduate School of Medical Sciences, University of Groningen.

## Author contributions

C.G.B. designed the hypothesis, performed the analyses, made the figures and tables, and wrote the paper. D.R. created the RS microbiome dataset, C.M.G. provided data analysis help, S.G. and A.K. performed replication analysis, D.S. provided and scored MRI data, P.A. and T.K. performed qPCR and analysis, A.M.I. contributed data of the RS cohort, S.B.Z. provided the MRI data, A.G.U. provided the gut-microbiome data of the Rotterdam Study, A.Z and J.F. contributed to collection of Lifelines-DEEP microbiome data and to the replication analysis data, R.K. created and provided the RS gut-microbiome dataset, J.B.J.v.M. designed the study and supervised this work. All authors critically assessed the paper.

## Competing interests

The authors declare no competing interests.
