## [Peer Review File · Nature Communications]

Reviewers' comments:

Reviewer #1 (Remarks to the Author):

This interesting study by Boer et al entitled 'Gut Microbiome Composition and its Relation to Joint Pain and Inflammation' provides evidence from the Rotterdam and LLD human cohorts that *Streptococcus* spp. or Streptococcaceae family members are associated with WOMAC score for knee pain. This association seems to hold up when controlling for covariates, including smoking, alcohol consumption, ethnicity, and medications that may impact *Streptococcus* spp. abundance (NSAIDs and PPIs). A correlation between *Streptococcus* spp. and knee joint effusion measured by MRI was also observed in a small subset of females in the Rotterdam cohort. Interestingly, when controlling for BMI, the association between *Streptococcus* spp. and WOMAC was lost. Key limitations to the study are mentioned by the authors, including that there is no evidence of a dysbiosis of the gut microbiome in knee OA in the cohorts studied, and there is no evidence that *Streptococcus* spp. is involved causally with either accelerated OA or joint pain. This second issue particularly reduces the impact and relevance of these findings, significantly reducing the suitability of the report for publication in Nat Comm.

Can the authors explain why when adjusted for BMI, the correlation with *Streptococcus* spp. is lost? Streptococcaceae family members have been implicated in driving systemic inflammation in obesity, and a lack of a correlation here is somewhat surprising.

Were any taxa lost or reduced in the cohort with evidence of knee pain (i.e. inversely associated with knee pain)? If so, that should be reported, and it could be

The Table 3 legend mentions two models, each adjusted differently. How are these models represented in the table? It seems like only data from one model is represented. Also, +WOMAC is mentioned for the second set of correlations, is the first set related to effusion score (not obvious)?

The authors report that there is no association between knee OA severity and gut microbiome. Diagnosis of knee OA involves structural assessments and not pain, and thus this report is not really about OA, but rather knee joint pain. Excluding OA from the equation, is there a dysbiosis (i.e. unique microbial community structure) that is associated with knee joint pain?

In the databases utilized in this study, is there any documentation of other joint pain (hip, upper extremity, hand)? Do these correlations restrict to knee pain, or other forms of joint pain? Is there any reason to think the correlation would be knee pain specific?

Is there a correlation between *Streptococcus* spp. and diagnosis of knee OA – or any form of OA – in the medical record?

The model Figure 2 is not appropriate as there is no evidence presented that any of the depicted molecular/metabolic pathways are involved in connecting *Streptococcus* spp. with the host.

Reviewer #2 (Remarks to the Author):

The association between gut microbiome and pathogenesis of osteoarthritis (OA) has been a hot topic in the field. Currently, Dr. Boer and her colleagues have demonstrated their finding of associations between increased abundance of *Streptococcus* spp. and increased knee pain in the Rotterdam cohort. This finding was then validated by stool samples collected from Lifelines Deep cohort. After that they also showed the association was driven by local inflammation in the knee joint by MRI data.

Overall the current study is of enough novelty though the connections between microbiome and pathogenesis of OA was speculated and verified by teams of Dr. Kraus^{1, 2} and Dr. Zuscik³. And the cohorts used in the current study were well known and recognized in the field of OA research. To my knowledge, this is the first study using large population of cohorts to looking for the association between gut microbiome and pathogenesis of OA. Some of the data were presented in the Osteoarthritis Research Society International (OARSI) 2017 world congress.

My concerns on the paper were as follows: (1) The authors have detailed explained the Rotterdam Study in the supplementary documents. However, as to the collection of stool samples, they did not explain in detail. Many studies have shown that microbiome profile would shift by different methods of stool sample collection, including temperature, time of transferring to -80°C etc.; (2) The main finding of the current study was the association between *Streptococcus* spp. and increased knee pain. However, association is not causation. The perfect microbiome related pathogenesis study is supposed to follow Koch's postulates. The authors have fully discussed the limitation of lacking causality in the paper. I still believe they are supposed to promote future studies established on

Koch's postulates; (3) As we know the Rotterdam study is a prospective longitudinal design. Does the authors have further follow-up data so they would have the possible to explore the association between gut microbiome and disease progression?; (4) Does the authors looking into the blood samples collected from both cohort so they can detect the vesicles as they proposed in the paper.

In summary, the current study is of enough novelty and workload. The writing and illustrations are well done. My suggestions on the paper is major revision by addressing the four questions raised above.

Thank you for inviting me to review this paper. If you have any questions, please don't hesitate to contact me.

References

1. ZeYu Huang, Virginia Kraus. Does lipopolysaccharide-mediated inflammation have a role in OA? *Nat Rev Rheumatol*. 2016; 12(2): 123-129.
2. ZeYu Huang, Stabler Thomas, FuXing Pei, Virginia Kraus. Both systemic and local lipopolysaccharide (LPS) burden are associated with knee OA severity and inflammation. *Osteoarthritis Cartilage*. 2016; 24(10): 1769-1775.
3. Schott, E. M., Farnsworth CW, Grier A, Lillis JA, Soniwala S, Dadourian GH, et al. Targeting the gut microbiome to treat the osteoarthritis of obesity. *JCI Insight*. 2018; 19; 3(8).

Reviewer #3 (Remarks to the Author):

Review of "Gut Microbiome Composition And Its Relation To Joint Pain and Inflammation," by Boer, et al.

In this article the authors use microbiome data from two studies (Rotterdam Study and Lifelines Deep Study) to look for associations between the fecal microbiome and arthritis pain in the knees. From this, the authors find an association between some type of Streptococcus and arthritis. I have several concerns about the papers.

First, the analyses of amplicon data need to be updated. Amplicon data should be run through a pipeline that establishes distinct sequence variants (such as DADA2, Deblur, UNOISE2, etc). Using a 3% OTU definition (i.e. using UPARSE at 97% identity) has been shown to be a poor method of taxonomy (see the papers for DADA2, etc; also Nguyen et al. *npj Biofilms and Microbiomes* 2: 16004 (2016)). Furthermore, downstream identification should be done using programs such as SPINGO that provide more accurate taxonomy, preferably down the species level if possible. Looking at the microbiome at such a high-level (e.g. phylum in Fig 1) is very uninformative.

Second, and similarly, the authors “aggregated OTUs at each taxonomical level if the taxonomic classification was identical.” Aggregation, and the manner in which it is applied, affects all downstream analysis, in particular those involving diversity. On the surface, this artificially changes the number of distinct groups which changes the estimate of diversity. At a deeper level, it becomes unclear what the groups actually represent and thus unclear what a diversity measure means (i.e., having some groups that represent genera or higher and others that might represent species makes little sense from a diversity standpoint).

Third, using PERMANOVA on Bray-Curtis dissimilarity is confusing (inappropriate?) because it doesn't obey Bray-Curtis does not obey the triangle property. This is a fancy way of saying that if A and B have some dissimilarity x , and B and C have dissimilarity y , there is no way to use the measures x and y to calculate the distance z between A and C that holds for all points in the dataset. The documentation on 'adonis' in 'vegan' acknowledges this issue (see the Notes section and the Details section about Bray-Curtis being “semimetric”). Statistically speaking, just because the mean Bray-Curtis dissimilarity is, e.g., linearly increasing with some measure doesn't mean the variance is scaling appropriately. The paper Warton, D. I., Wright, S. T., & Wang, Y. (2011). Distance-based multivariate analyses confound location and dispersion effects. *Methods in Ecology and Evolution*, 3(1), 89–101 referenced in the vegan package explains many of the issues. The aforementioned paper recommends ways of circumventing the issue (they suggest using 'mvabund' with appropriate error terms, see page 97). Note that Warton et al. found that using PERMANOVA + Bray-Curtis typically only detects effects for high variance taxa.

Fourth, the authors use a package MaAsLin which I am not terribly familiar with. However, it looks as if it performs boosted regression coupled with regularization. The authors state that they use arcsine-square root (ASR) transformation of the abundance table for these analyses. Work on the analysis of compositional data, like microbial communities represented by relative abundance, has shown that ASR does not remove collinearity and that transforms like the isometric log-ratio transform perform better (Aitchison, J. (1986) *The Statistical Analysis of Compositional Data*, Monographs on Statistics and Applied Probability. Chapman & Hall Ltd., London (UK) followed up as well in works by Egozcue, Pawłowsky-Glahn et al.). Using boosted regression (or regularization) is a

reasonable approach however probably not necessary given the small number of covariates being tested, and it might be more advantageous to more flexible regression packages than MaAsLin.

Finally, I have comment which I don't really see how the authors can fix. They have single time point from each subject. We know from many studies that (1) the gut microbiome changes fairly rapidly and (2) environmental drivers affect the gut microbiome (e.g. what you ate yesterday). It seems very hard to take something which is temporally highly dynamic and then compare it to a long-term chronic condition such as osteoarthritis. It seems like you would need at least some time series data showing that across some period time (a month? A year?) some bacterial species is on-average the dominant type. Can we really hope to extrapolate from one data point a chronic condition? Particularly given the uniqueness of host genetics, environment, etc that all play a role in microbiome composition.

As a minor comment, there were quite a few typos which I found but will not detail given the larger issues with the paper. Another minor comment is that, it has been shown by several studies, that the fecal microbiome is not the same as the gut microbiome. It would be wise to temper the language of the manuscript in light of this fact.

Given the problems with analyses I am not convinced of what the authors are presenting. Here is what I think you probably can conclude: the rather amorphous group which has been given the moniker "Streptococcus" was highly variable in subjects in both studies.

General comment to all reviewers

We thank the reviewers for their comments which have helped to improve our manuscript. All references to page and line numbers correspond to page and line numbers in the .pdf version of our revised manuscript.

Reviewer #1 (Remarks to the Author):

This interesting study by Boer et al entitled 'Gut Microbiome Composition and its Relation to Joint Pain and Inflammation' provides evidence from the Rotterdam and LLD human cohorts that *Streptococcus* spp. or Streptococcaceae family members are associated with WOMAC score for knee pain. This association seems to hold up when controlling for covariates, including smoking, alcohol consumption, ethnicity, and medications that may impact *Streptococcus* spp. abundance (NSAIDs and PPIs). A correlation between *Streptococcus* spp. and knee joint effusion measured by MRI was also observed in a small subset of females in the Rotterdam cohort. Interestingly, when controlling for BMI, the association between *Streptococcus* spp. and WOMAC was lost.

- 1. Key limitations to the study are mentioned by the authors, including that there is no evidence of a dysbiosis of the gut microbiome in knee OA in the cohorts studied, and there is no evidence that *Streptococcus* spp. is involved causally with either accelerated OA or joint pain. This second issue particularly reduces the impact and relevance of these findings, significantly reducing the suitability of the report for publication in Nat Comm.**

Reply: We have reported an association of gut microbiome composition with OA-specific knee joint pain. Indeed, lack of evidence for causation is a limitation of our study, as we discuss in the manuscript (*see discussion page 17*). Although each study should stand on its own, to our knowledge, in general, high profile published microbiome association studies did not present evidence for causality; did not have a second time point, did not validate their sequencing method, or did not replicate findings in other cohorts¹⁻⁵. We validated and replicated our *Streptococcus* spp. association indicating this to be a robust finding.

We do not find a statistically significant dysbiosis (β -diversity) of the gut microbiome with knee OA severity ($p=0.09$, $R^2=0.00139$). It is very likely that we currently lack the power to detect such an association, if it exists or that we need to study this aspect in a longitudinal manner. (*see also point 5 of reviewer 1*). We have added some extra lines of discussion (*pdf: page 21 lines 388-344*)

- 2. Can the authors explain why when adjusted for BMI, the correlation with *Streptococcus* spp. is lost? Streptococcaceae family members have been implicated in driving systemic inflammation in obesity, and a lack of a correlation here is somewhat surprising.**

Reply: In contrast to what the reviewer states, the association between *Streptococcae* and knee pain was not lost after BMI adjustment. When we correct for BMI the association is slightly attenuated (*Streptococcus* coefficient= 5.04×10^{-03} , bmi-adjusted= 4.13×10^{-03}), but it remains significant (**results page 9 lines 142-144, supplementary table S3**). This indicates that only part of the association between *Streptococcus* and pain is explained by BMI. In the meta-analysis the association with *Streptococcus* is robust and highly significant in the analysis adjusted for BMI (**supplementary table S9**).

3. Were any taxa lost or reduced in the cohort with evidence of knee pain (i.e. inversely associated with knee pain)? If so, that should be reported, and it could be

Reply: No taxa were lost (*see results page 9 lines 132-138*). For completion, we have added the full taxa analysis for the basic model (adjusted for age, sex and technical covariates) to our results in **supplementary table S2**. Overall, we only see an association with the clade of *Streptococcus spp.*

4. The table 3 legend mentions two models, each adjusted differently. How are these models represented in the table? It seems like only data from one model is represented. Also, +WOMAC is mentioned for the second set of correlations, is the first set related to effusion score (not obvious)?

Reply : We apologize for the confusion in table 3, the legend wrongly described the models. We have now corrected the legend to accurately describe the models (**see table 3**).

5. The authors report that there is no association between knee OA severity and gut microbiome. Diagnosis of knee OA involves structural assessments and not pain, and thus this report is not really about OA, but rather knee joint pain. Excluding OA from the equation, is there a dysbiosis (i.e. unique microbial community structure) that is associated with knee joint pain?

Reply: The diagnosis of OA can be defined in different ways: structural changes, and/or on clinical symptoms. This is a field of active discussion. The reviewer is correct that we here report an association specific with knee joint pain, and not with structural OA severity (**reported in results Page 8**). Nevertheless, the used pain questionnaire (Western Ontario MacMaster (WOMAC)) is a validated pain-score designed specifically for the assessment of lower extremity pain and function for OA of the knee. To clarify that we have examined an osteoarthritis specific joint pain score and not a general knee pain score, we have changed all mentions of knee pain to knee WOMAC-pain score. Furthermore, we have added sentences to the introduction to explain this better (**Introduction, page 4, lines 73-76 and Methods, page 27, lines 447 - 453**).

With the WOMAC-pain score we do indeed see a dysbiosis of the microbiome (**supplementary figure 1**). This dysbiosis is driven by *Streptococcus spp.* abundance (**table 2**).

6. In the databases utilized in this study, is there any documentation of other joint pain (hip, upper extremity, hand)? Do these correlations restrict to knee pain, or other forms of joint pain? Is there any reason to think the correlation would be knee pain specific?

Reply: The Rotterdam Study dataset has questionnaire data regarding the presence of joint pain (yes/no) for hip and hand as well. However, these data are different compared to the WOMAC-pain score, since this is only 1 question about presence/absence of pain, and the severity of pain/complaints is not included. Consequently, power to detect a robust associations is small when using these dichotomous pain phenotypes. In addition, there is evidence that inflammation plays a more important role in OA in the knee joint as compared to the hip joint⁶⁻⁹. However, it is plausible that similar mechanisms might occur in other inflammatory joint disorders and/or other pain measurements. We have added a paragraph about this in the discussion (**discussion, page 19, lines 289-297 and Discussion, page 21, lines 330-333**).

Is there a correlation between *Streptococcus* spp. and diagnosis of knee OA – or any form of OA – in the medical record?

Reply : We actually studied the association between structural knee OA and the microbiome, which was reported in the **results, page 9 lines 127- 128**. However, power to detect an association was low due to the limited number of cases in this relatively young population.

There is literature reporting the association of *Streptococcus* infections with other rheumatic diseases, specifically reactive arthritis and rheumatic fever¹⁰⁻¹³. In addition, *Streptococcus* is also associated with osteomyelitis^{14,15}. These studies are part of the discussion of our manuscript (**Discussion, Page 19 lines 291-300**).

The model Figure 2 is not appropriate as there is no evidence presented that any of the depicted molecular/metabolic pathways are involved in connecting *Streptococcus* spp. with the host.

Reply: Indeed, **figure 2** is entirely a hypothetical proposed model for the possible causality between *Streptococcus* spp. and WOMAC- pain score. The illustration serves solely to provide clarity to the written description: (1) to illustrate the biological plausibility of our finding and (2) to suggest possible follow-up experiments that can confirm our hypothesis (**Discussion page 20-21, lines 324-330**) The figure is of importance for textual understanding, however it must not be misinterpreted as anything other than purely hypothetical. Therefore, we have adapted the figure and placed emphasis on the hypothetical nature of the figure (see **Discussion, page 20, lines 310-319** and **Figure 2 legend**). It is however possible to delete the figure if required.

Reviewer #2 (Remarks to the Author):

The association between gut microbiome and pathogenesis of osteoarthritis (OA) has been a hot topic in the field. Currently, Dr. Boer and her colleagues have demonstrated their finding of associations between increased abundance of *Streptococcus* spp. and increased knee pain in the Rotterdam cohort. This finding was then validated by stool samples collected from Lifelines Deep cohort. After that they also showed the association was driven by local inflammation in the knee joint by MRI data.

Overall the current study is of enough novelty though the connections between microbiome and pathogenesis of OA was speculated and verified by teams of Dr. Kraus^{1, 2} and Dr. Zuscik³. And the cohorts used in the current study were well known and recognized in the field of OA research. To my knowledge, this is the first study using large population of cohorts to looking for the association between gut microbiome and pathogenesis of OA. Some of the data were presented in the Osteoarthritis Research Society International (OARSI) 2017 world congress.

My concerns on the paper were as follows:

1. The authors have detailed explained the Rotterdam Study in the supplementary documents. However, as to the collection of stool samples, they did not explain in detail. Many studies have shown that microbiome profile would shift by different methods of stool sample collection, including temperature, time of transferring to -80°C etc.

Reply : We have provided information on sample collection in more detail (see **Methods page 24-27, lines 394 - 445**). The methods for collection of the replication study (LifeLines study) have been published before, and are referenced in the manuscript (**Methods, page 28, lines 471 – 492,, reference nr: 27 and 53**).

2. The main finding of the current study was the association between *Streptococcus* spp. and increased knee pain. However, association is not causation. The perfect microbiome related pathogenesis study is supposed to follow Koch's postulates. The authors have fully discussed the limitation of lacking causality in the paper. I still believe they are supposed to promote future studies established on Koch's postulates;

Reply: As noted by the reviewer we have addressed this explicitly in the discussion. We now have added possible future studies based on epidemiological criteria for causation (Koch and Bradford-Hill)¹⁶. This is now included in the discussion (**Discussion, page 20, lines 320 - 344**).

3. As we known the Rotterdam study is a prospective longitudinal design. Does the authors have further follow-up data so they would have the possible to explore the association between gut microbiome and disease progression?;

Reply: Unfortunately, the only stool samples collected and analysed up to now, are from a recent visit of the youngest Rotterdam study cohort, RS-III, between 2012-2014. Presently, we have no longitudinal data available for analysis. Nevertheless, we are currently gathering follow-up data for osteoarthritis progression, which we are eager to analyse in the future. We have added this suggestion to the discussion (**Discussion, page 20, lines 320-344**).

- 4. Does the authors looking into the blood samples collected from both cohort so they can detect the vesicles as they proposed in the paper.**

Reply: This is a very good suggestion of the reviewer , however we currently do not have such data but we are planning exactly these experiments. We have added this suggestion to discussion on *(Discussion, page 20, lines 320-344)*

In summary, the current study is of enough novelty and workload. The writing and illustrations are well done. My suggestions on the paper is major revision by addressing the four questions raised above.

Reviewer #3 (Remarks to the Author):

Review of “Gut Microbiome Composition And Its Relation To Joint Pain and Inflammation,” by Boer, et al. In this article the authors use microbiome data from two studies (Rotterdam Study and Lifelines Deep Study) to look for associations between the fecal microbiome and arthritis pain in the knees. From this, the authors find an association between some type of Streptococcus and arthritis. I have several concerns about the papers.

1. **First, the analyses of amplicon data need to be updated. Amplicon data should be run through a pipeline that establishes distinct sequence variants (such as DADA2, Deblur, UNOISE2, etc). Using a 3% OTU definition (i.e. using UPARSE at 97% identity) has been shown to be a poor method of taxonomy (see the papers for DADA2, etc; also Nguyen et al. *npj Biofilms and Microbiomes* 2: 16004 (2016)).**

Reply: We have used the 3% OTU definition, as it is considered as one of the “golden standard” methodologies for 16S rRNA studies, as demonstrated by its use in the Human Microbiome Project (<https://www.hmpdacc.org/HMQCP/>). Also, it is the standard setting of the well-known and widely used microbiome analysis pipeline QIIME, which has been cited/used in many (over 14000) papers¹⁷. QIIME, is still widely in use, (please see a number of recent high impact publications using QIIME^{1-3,18-20}). In addition, a paper from 2017 compared the use of QIIME, UPARSE and DADA2 concluded for QIIME and DADA2 that “*both bioinformatic approaches support essentially the same biological conclusions*” and “*..samples from variable treatment groups were differentiated from one another regardless of the sequencing platform and/or bioinformatics pipeline used allowing us to draw similar conclusions*”²¹.

We have altered our amplicon analysis data, by using direct read classification using RDP, please see our response to point 3 made by reviewer 3.

2. **Furthermore, downstream identification should be done using programs such as SPINGO that provide more accurate taxonomy, preferably down the species level if possible. Looking at the microbiome at such a high-level (e.g. phylum in Fig 1) is very uninformative.**

Reply: Indeed, taxonomy classification down to the species level would be strongly preferable, however, it is not possible to reliably call taxonomies down to the species level with our methods. Although SPINGO does claim to provide taxonomy to species level, UPARSE/QIIME generally outperform SPINGO^{22,23}.

Regarding figure 1, it merely depicts an overview of the composition of the Rotterdam Study gastrointestinal microbiome composition and is similar in nature to other papers of large population microbiome cohorts^{5,24,25}. This is done so that the reader can compare the microbiome composition of the Rotterdam Study to that of previously published data. All of our analysis are focused on the lowest taxonomic level ,i.e., genus. We show results from higher taxonomies to illustrate the strength and consistency of the association.

We have altered our taxonomic classification method, please see our response to point 3 made by reviewer3.

- 3. Second, and similarly, the authors “aggregated OTUs at each taxonomical level if the taxonomic classification was identical.” Aggregation, and the manner in which it is applied, affects all downstream analysis, in particular those involving diversity. On the surface, this artificially changes the number of distinct groups which changes the estimate of diversity. At a deeper level, it becomes unclear what the groups actually represent and thus unclear what a diversity measure means (i.e., having some groups that represent genera or higher and others that might represent species makes little sense from a diversity standpoint).**

Reply : We feel that the reviewer might have misunderstood what we meant with the aggregation of OTU's at each taxonomic level. For both, the alpha- and β -diversity analysis, we had used the full OTU table (thus all OTU's identified in the open reference calling analysis). The grouping that we had done was after taxonomic classification only for the MaAslin analysis and was purely to reduce the multiple testing burden.

Because of the concerns described by the reviewer (also in point 1 and point 2), we have now done all our analysis using a “closed-reference” based method. In this new version of the manuscript, we have directly classified reads using RDP-classifier to the SILVA 16S sequence database, also known as “closed reference calling”²⁶. We have done this separately for each studied taxonomic level, with a binning posterior probability cut-off of 0.8 (see methods, page 26, lines 422 - 443). Thus, no aggregation or grouping was necessary or done (see also Methods, statistical analysis, page 30, lines 525 – 573). As can be seen from all our results (tables 2-3, supplementary tables 3-9, and results), the reported association of *Streptococcus spp.* with WOMAC-pain score remains highly significant and robust.

- 4. Third, using PERMANOVA on Bray-Curtis dissimilarity is confusing (inappropriate?) because it doesn't obey Bray-Curtis does not obey the triangle property. This is a fancy way of saying that if A and B have some dissimilarity x, and B and C have dissimilarity y, there is no way to use the measures x and y to calculate the distance z between A and C that holds for all points in the dataset. The documentation on 'adonis' in 'vegan' acknowledges this issue (see the Notes section and the Details section about Bray-Curtis being “semimetric”). Statistically speaking, just because the mean Bray-Curtis dissimilarity is, e.g., linearly increasing with some measure doesn't mean the variance is scaling appropriately. The paper Warton, D. I., Wright, S. T., & Wang, Y. (2011). Distance-based multivariate analyses confound location and dispersion effects. *Methods in Ecology and Evolution*, 3(1), 89–101 referenced in the vegan package explains many of the issues. The aforementioned paper recommends ways of circumventing the issue (they suggest using 'mvabund' with appropriate error terms, see page 97). Note that Warton et al. found that using PERMOVA + Bray-Curtis typically only detects effects for high variance taxa.**

Reply: We have used the Bray-Curtis β -diversity and subsequent PERMANOVA analysis, as it is commonly used in (population-based) microbiome analysis^{3-5,27,28}.

We did run the mvabund analysis, as recommended by the reviewer and the results also show a highly significant association for microbiome diversity and WOMAC-pain. (model results are depicted below)

Model used: `microbiome.genera ~ sex + age + TimeinMail + Batch + WOMAC-pain score`

Family used: "negative binomial"
Multivariate test results:

	Res.Df	Df.diff	Dev	Pr(>Dev)
Sex	1396	1	3283	0.001 ***
Age	1395	1	2434	0.001 ***
TimeInMail	1394	1	1858	0.001 ***
Batch	1393	1	3138	0.001 ***
WOMAC-pain	1392	1	1697	0.001 ***

Signif. codes: 0 '***' 0.001 '**' 0.01 '*' 0.05 '.' 0.1 ' ' 1

Arguments: Test statistics calculated assuming uncorrelated response (for faster computation). P-value calculated using 999 resampling iterations via PIT-trap resampling (to account for correlation in testing).

However, a recent report argued that other analysis methods might indeed be better to use for large population compositional microbiome data analysis²⁹. This method is based on CLR (Centred Log-Ratio) normalization, Aitchison distances and then PERMANOVA analysis. This is the method we now report in our manuscript, see **Methods, statistical analysis, page 30, lines 525 – 573, Results page 8, lines 119–123, Supplementary table S1 and Supplementary figure 1**. We have chosen the CLR method since it was highly recommended for large population based analysis²⁹. In addition, the mvabund analysis was very computational intensive and seems to be more compatible for smaller sized studies. We will leave it to the discretion of the editor to also include the mvabund analysis results in the supplementary materials.

5. **Fourth, the authors use a package MaAsLin which I am not terribly familiar with. However, it looks as if it performs boosted regression coupled with regularization. The authors state that they use arcsine-square root (ASR) transformation of the abundance table for these analyses. Work on the analysis of compositional data, like microbial communities represented by relative abundance, has shown that ASR does not remove collinearity and that transforms like the isometric log-ratio transform perform better (Aitchison, J. (1986) *The Statistical Analysis of Compositional Data, Monographs on Statistics and Applied Probability*. Chapman & Hall Ltd., London (UK) followed up as well in works by Egozcue, Pawlowsky-Glahn et al.). Using boosted regression (or regularization) is a reasonable approach however probably not necessary given the small number of covariates being tested, and it might be more advantageous to more flexible regression packages than MaAsLin.**

Reply: We have followed the suggestion of the reviewer and also performed the association analysis after an isometric log-ratio (ISR) transformation on our data. We have now added these results in **supplementary table S7**, and described this in our **methods (Methods, statistical analysis, page 31, lines 560-564)** and in the **results (Page 14, lines 206 - 216)**. Overall, the association between *Streptococcus* and WOMAC score, remains robust after transformation (CoE = 7.01×10^{-02} , $p = 8.35 \times 10^{-04}$). These results are also in line with the results obtained by determining the absolute *Streptococcus* abundance (as obtained by qPCR) in relation to the WOMAC score.

6. **Finally, I have comment which I don't really see how the authors can fix. They have single time point from each subject. We know from many studies that (1) the gut microbiome changes fairly rapidly and (2) environmental drivers affect the gut microbiome (e.g. what**

you ate yesterday). It seems very hard to take something which is temporally highly dynamic and then compare it to a long-term chronic condition such as osteoarthritis. It seems like you would need at least some time series data showing that across some period time (a month? A year?) some bacterial species is on-average the dominant type. Can we really hope to extrapolate from one data point a chronic condition? Particularly given the uniqueness of host genetics, environment, etc that all play a role in microbiome composition.

Reply: Indeed extra data time points would be preferred to establish causality (we have addressed this issue in our discussion) However, currently no such data is available

Host genetics only explains a very small part of the host microbiome composition; ethnicity (cultural diet) and environment are more prominent determinants³⁰. Our cohort consists solely of individuals from the same geographical area in Rotterdam, the same socio-economic class and age range³¹, reducing confounding factors of the environment. Also, we have adjusted for alcohol and smoking use, further reducing environmental effects (**results, page 6, lines 139-141, table S3**). Our association is not due to environmental factors of our cohort, since we have replicated our findings in an independent cohort (LifeLines-DEEP), whose participants come from a different geographical area, have different age ranges and different and socio-economic classes as our cohort³². As we describe in our manuscript (**results, page 15, lines 218-228, supplementary table S8**), ethnicity has recently been put forward as a possible confounder for gastrointestinal microbiome composition. When we excluded individuals with non-Caucasian ethnicity, our association remained significant (**Supplementary table S4**). Indicating, that our association is also not driven by ethnicity.

Regarding the variability of the human gut microbiome, fast changes in the microbiome have been observed with dietary interventions (shifts from meat to plant based diet and vice versa)³³, however, this is not observed with all dietary interventions³⁴. Furthermore, long term microbiome research shows that the gut microbiome is highly stable over long periods of time^{35,36}, as is the diet in the general population. Altogether, subtle fluctuations are seen, and can be introduced by dietary changes, however the microbial community also rapidly returns to its stable state. Finding an average, 60% of bacterial strains remained stable for up to 5 years and many were estimated to remain stable for decades³⁵. In sum, these studies and others support and indicate the suitability of the gut microbiome as a diagnostic tool and therapeutic target.

7. As a minor comment, there were quite a few typos which I found but will not detail given the larger issues with the paper.

Reply: To the best of our abilities we have proofread the new version of our manuscript.

8. Another minor comment is that, it has been shown by several studies, that the faecal microbiome is not the same as the gut microbiome. It would be wise to temper the language of the manuscript in light of this fact

Reply: We have updated the manuscript to more clearly reflect that the stool microbiome is only an approximation of the gut microbiome (**see introduction, page 4, lines 73 -75, Results page 5, lines 83-86**).

9. Given the problems with analyses I am not convinced of what the authors are presenting. Here is what I think you probably can conclude: the rather amorphous group which has been given the moniker “Streptococcus” was highly variable in subjects in both studies.

Reply: We have tried to follow the suggestions of the reviewer and re-analysed the data following the suggested alternative statistical approaches, obtaining similar results. Besides these analysis, we want to call the attention of the reviewer to the fact that we also validated our finding by absolute quantification of *Streptococcus spp.* by qPCR, which can be considered the “gold standard” for quantification of taxonomical groups in stool microbiome³⁸. Furthermore, our findings were replicated in an independent cohort, which used different collection, sequencing and data processing protocols.

We believe the key strength of our study is the use of a large collection of samples. For small scale analysis, differences in statistical methods could indeed have a large impact on the results and interpretation of the findings. However, results coming from large datasets are in general more robust as it is observed in related fields of analysis of *-omics* data.

References:

1. Consortium, T. H. M. P. *et al.* Structure, function and diversity of the healthy human microbiome. *Nature* **486**, 207–214 (2012).
2. Hansen, M. E. B. *et al.* Population structure of human gut bacteria in a diverse cohort from rural Tanzania and Botswana. *Genome Biol.* **20**, 16 (2019).
3. Zhernakova, A. *et al.* Population-based metagenomics analysis reveals markers for gut microbiome composition and diversity. *Science (80-.)*. **352**, 565–569 (2016).
4. Deschasaux, M. *et al.* Depicting the composition of gut microbiota in a population with varied ethnic origins but shared geography. *Nat. Med.* **24**, 1526–1531 (2018).
5. Dubois, G., Girard, C., Lapointe, F.-J. & Shapiro, B. J. The Inuit gut microbiome is dynamic over time and shaped by traditional foods. *Microbiome* **5**, 151 (2017).
6. Kraus, V. B. *et al.* Direct in vivo evidence of activated macrophages in human osteoarthritis. *Osteoarthr. Cartil.* **24**, 1613–1621 (2016).
7. Reyes, C. *et al.* Association Between Overweight and Obesity and Risk of Clinically Diagnosed Knee, Hip, and Hand Osteoarthritis: A Population-Based Cohort Study. *Arthritis Rheumatol. (Hoboken, N.J.)* **68**, 1869–75 (2016).
8. Utomo, L., van Osch, G. J. V. M., Bayon, Y., Verhaar, J. A. N. & Bastiaansen-Jenniskens, Y. M. Guiding synovial inflammation by macrophage phenotype modulation: an in vitro study towards a therapy for osteoarthritis. *Osteoarthr. Cartil.* **24**, 1629–38 (2016).
9. Huang, Z. Y., Stabler, T., Pei, F. X. & Kraus, V. B. Both systemic and local lipopolysaccharide (LPS) burden are associated with knee OA severity and inflammation. *Osteoarthr. Cartil.* **24**, 1769–1775 (2016).
10. Mackie, S. L. & Keat, A. Poststreptococcal reactive arthritis: what is it and how do we know? *Rheumatology* **43**, 949–954 (2004).
11. Tandon, R. *et al.* Revisiting the pathogenesis of rheumatic fever and carditis. *Nat. Rev. Cardiol.* **10**, 171–177 (2013).
12. Cunningham, M. W. Rheumatic Fever, Autoimmunity, and Molecular Mimicry: The Streptococcal Connection. *Int. Rev. Immunol.* **33**, 314–329 (2014).
13. Barash, J. Rheumatic Fever and Post-Group A Streptococcal Arthritis in Children. *Curr. Infect. Dis. Rep.* **15**, 263–268 (2013).
14. Murillo, O. *et al.* Streptococcal vertebral osteomyelitis: multiple faces of the same disease. *Clin. Microbiol. Infect.* **20**, O33–O38 (2014).
15. McGuire, T., Gerjarusak, P., Hinthorn, D. R. & Liu, C. Osteomyelitis caused by beta-hemolytic streptococcus group B. *JAMA* **238**,

- 2054–5 (1977).
16. Fedak, K. M., Bernal, A., Capshaw, Z. A. & Gross, S. Applying the Bradford Hill criteria in the 21st century: how data integration has changed causal inference in molecular epidemiology. *Emerg. Themes Epidemiol.* **12**, 14 (2015).
 17. Caporaso, J. G. *et al.* QIIME allows analysis of high-throughput community sequencing data. *Nat. Methods* **7**, 335 (2010).
 18. Strandwitz, P. *et al.* GABA-modulating bacteria of the human gut microbiota. *Nat. Microbiol.* **4**, 396–403 (2019).
 19. Nakamoto, N. *et al.* Gut pathobionts underlie intestinal barrier dysfunction and liver T helper 17 cell immune response in primary sclerosing cholangitis. *Nat. Microbiol.* **4**, 492–503 (2019).
 20. Jin, C. *et al.* Commensal Microbiota Promote Lung Cancer Development via $\gamma\delta$ T Cells. *Cell* **176**, 998–1013.e16 (2019).
 21. Allali, I. *et al.* A comparison of sequencing platforms and bioinformatics pipelines for compositional analysis of the gut microbiome. *BMC Microbiol.* **17**, 194 (2017).
 22. Edgar, R. C. Accuracy of taxonomy prediction for 16S rRNA and fungal ITS sequences. *PeerJ* **6**, e4652 (2018).
 23. Escobar-Zepeda, A. *et al.* Analysis of sequencing strategies and tools for taxonomic annotation: Defining standards for progressive metagenomics. *Sci. Rep.* **8**, (2018).
 24. Falony, G. *et al.* Population-level analysis of gut microbiome variation. *Science (80-.)*. **352**, 560–564 (2016).
 25. Hansen, M. E. B. *et al.* Population structure of human gut bacteria in a diverse cohort from rural Tanzania and Botswana. *Genome Biol.* **20**, 16 (2019).
 26. Wang, Q., Garrity, G. M., Tiedje, J. M. & Cole, J. R. Naive Bayesian classifier for rapid assignment of rRNA sequences into the new bacterial taxonomy. *Appl. Environ. Microbiol.* **73**, 5261–7 (2007).
 27. McKnight, D. T. *et al.* Methods for normalizing microbiome data: An ecological perspective. *Methods Ecol. Evol.* **10**, 389–400 (2019).
 28. Ross, A. A., Müller, K. M., Weese, J. S. & Neufeld, J. D. Comprehensive skin microbiome analysis reveals the uniqueness of human skin and evidence for phyllosymbiosis within the class Mammalia. *Proc. Natl. Acad. Sci.* **115**, E5786–E5795 (2018).
 29. Gloor, G. B., Macklaim, J. M., Pawlowsky-Glahn, V. & Egozcue, J. J. Microbiome Datasets Are Compositional: And This Is Not Optional. *Front. Microbiol.* **8**, 2224 (2017).
 30. Rothschild, D. *et al.* Environment dominates over host genetics in shaping human gut microbiota. *Nature* **555**, 210–215 (2018).
 31. Ikram, M. A. *et al.* The Rotterdam Study: 2018 update on objectives, design and main results. *Eur. J. Epidemiol.* **32**, 807–850 (2017).
 32. Tigchelaar, E. F. *et al.* Cohort profile: LifeLines DEEP, a prospective, general population cohort study in the northern Netherlands: study design and baseline characteristics. *BMJ Open* **5**, e006772 (2015).
 33. David, L. A. *et al.* Diet rapidly and reproducibly alters the human gut microbiome. *Nature* **505**, 559–63 (2014).
 34. Roager, H. M. *et al.* Whole grain-rich diet reduces body weight and systemic low-grade inflammation without inducing major changes of the gut microbiome: a randomised cross-over trial. *Gut* **68**, 83–93 (2019).
 35. Faith, J. J. *et al.* The Long-Term Stability of the Human Gut Microbiota. *Science (80-.)*. **341**, 1237439 (2013).
 36. de Meij, T. G. J. *et al.* Composition and stability of intestinal microbiota of healthy children within a Dutch population. *FASEB J.* **30**, 1512–1522 (2016).
 37. Sanna, S. *et al.* Causal relationships among the gut microbiome, short-chain fatty acids and metabolic diseases. *Nat. Genet.* **51**, 600–605 (2019).
 38. Vandeputte, D. *et al.* Quantitative microbiome profiling links gut community variation to microbial load. *Nature* **551**, 507–511 (2017).

Reviewer #1 (Remarks to the Author):

Very comprehensive and responsive revision.

Reviewer #2 (Remarks to the Author):

Dear Authors:

You have done a good job. I believe the authors have addressed all my questions. I have no more concerns and questions. I think it is publishable at current version.

Sincerely yours,

ZeYu Huang

Department of Orthopaedic Surgery

West China Hospital

SiChuan University

Reviewer #3 (Remarks to the Author):

I appreciate the authors' efforts to address the reviewers' concerns with their work. With particular reference to my review: the use Aitchison distance (instead of Bray-Curtis), alternate statistics, and a slightly modified classification scheme makes me feel more confident in their findings. I do take a bit of issue with the authors' "lemming" defense of using programs such as QIIME and UPARSE. Just because the Human Microbiome Project once advocated in its infancy (i.e. well over a decade ago) some method, or that X number of people have used a method is not a valid reason for refusal of newer methodologies. Recalcitrance to the uptake of new and improved algorithms and methods hinders scientific progress, particularly when it has been demonstrated that old methods are less informative or reliable. This is a large problem for the field of microbiome research. (Thus ends my soap-box speech.)

With regards to the latest version, I have a bit of an issue with some wording. The first is exemplified by this statement: "... overall microbiome composition (beta-diversity) was significantly associated

with knee WOMAC scores." Microbiome composition (a property of an individual) is not the same as beta-diversity (a property of 2 or more microbiome samples). Really what the authors have is the pairwise distance between 2 compositions compared against permuted contrasts of WOMAC scores. (Which it is also debatable whether pairwise is sufficient to establish beta-diversity, but it seems to be in fashion these days to claim 2 points is good enough to say something about variance between groups...) The reality of what has been measured and analyzed is quite a bit different from claiming that WOMAC score is affected by some individual property. Second, in both the paper and rebuttal letter, the term "dysbiosis" is used. Dysbiosis would suggest that the microbial community is harmful to a host in some way. I think that's a pretty strong overreach for finding a correlation between a genus of bacteria in stool samples and a knee-pain index. Correlation and causation are different things (as pointed out by the other reviewers) particularly for a complex, poorly defined trait such as pain. While the findings might hint at a dysbiosis, actually calling it that seems inappropriate.

Finally, I still worry that a single time point of observation is misleading to study when it comes to a chronic condition that may have taken years to develop. The other reviewers also pointed out that longitudinal data would greatly strengthen the findings of the paper. While I realize the data may not be there to do this, I think this is a huge weakness for the study.

Point by Point response to the reviewers 'comments:

On behalf of all co-authors we thank all reviewers for their review and comments.

Reviewer #3 (Remarks to the Author):

I appreciate the authors' efforts to address the reviewers' concerns with their work. With particular reference to my review: the use Aitchison distance (instead of Bray-Curtis), alternate statistics, and a slightly modified classification scheme makes me feel more confident in their findings. I do take a bit of issue with the authors' "lemming" defence of using programs such as QIIME and UPARSE. Just because the Human Microbiome Project once advocated in its infancy (i.e. well over a decade ago) some method, or that X number of people have used a method is not a valid reason for refusal of newer methodologies. Recalcitrance to the uptake of new and improved algorithms and methods hinders scientific progress, particularly when it has been demonstrated that old methods are less informative or reliable. This is a large problem for the field of microbiome research. (Thus ends my soap-box speech.)

A: We agree with the reviewer that the "lemming" defence is on its own indeed not a good argument to maintain the use of QIIME, and therefore not our main argument to use QIIME. We have chosen QIIME, as this method is still currently one of the best performing methods, as is demonstrated in these recent comparison articles¹⁻³. QIIME, performs equally good or better than the newer methods, such as DADA2 and SPINGO. We are currently collection new samples in our population cohort, and for our future data set we are most definably also looking at newly developed tools, as we agree that the field must not shy away from improvement and progress.

1. Allali, I. et al. A comparison of sequencing platforms and bioinformatics pipelines for compositional analysis of the gut microbiome. BMC Microbiol. 17, 194 (2017).
2. Edgar, R. C. Accuracy of taxonomy prediction for 16S rRNA and fungal ITS sequences. PeerJ 6, e4652 (2018).
3. Escobar-Zepeda, A. et al. Analysis of sequencing strategies and tools for taxonomic annotation: Defining standards for progressive metagenomics. Sci. Rep. 8, (2018).

With regards to the latest version, I have a bit of an issue with some wording. The first is exemplified by this statement: "... overall microbiome composition (beta-diversity) was significantly associated with knee WOMAC scores." Microbiome composition (a property of an individual) is not the same beta-diversity (a property of 2 or more microbiome samples). Really what the authors have is the pairwise distance to between 2 compositions compared against permuted contrasts of WOMAC scores. (Which it is also debatable whether pairwise is sufficient to establish beta-diversity, but it seems to be in fashion these days to claim 2 points is good enough to say something about variance between groups...) The reality of what has

been measured and analyzed is quite a bit different from claiming that WOMAC score is affected by some individual property. Second, in both the paper and rebuttal letter, the term "dysbiosis" is used. Dybiosis would suggest that the microbial community is harmful

to a host in some way. I think that's a pretty strong overreach for finding a correlation between a genus of bacteria in stool samples and a knee-pain index. Correlation and causation are different things (as pointed out by the other reviewers) particularly for a complex, poorly defined trait such as pain. While the findings might hint at a dysbiosis, actually calling it that seems inappropriate.

A: As requested by the reviewer we have changed the wording to “ We found that knee WOMAC-pain significantly contributes to the intestinal microbiome β -diversity as evaluated at genus level “, page 6 lines 106-108, (page 11, 235-236, page 13, lines 227-229). We agree with the reviewer that the use of “dysbiosis’ is a too strong of a statement, thus we have removed this wording from our article(page 11, 235-236, page 13, lines 227-229).

Finally, I still worry that a single time point of observation is misleading to study when it comes to a chronic condition that may have taken years to develop. The other reviewers also pointed out that longitudinal data would greatly strengthen the findings of the paper. While I realize the data may not be there to do this, I think this is a huge weakness for the study.

We acknowledge that this is our study’s greatest weakness (page 13, 280-282), and thus we ourselves are actively collecting additional samples at a second time point for future work and promoting follow-up research in our paper (page 13, 281-289)